# Effects of Serotonin on Cell Viability, Permeability of Bovine Mammary Gland Epithelial Cells and Their Transcriptome Analysis

**DOI:** 10.3390/ijms241411388

**Published:** 2023-07-13

**Authors:** Jie Lu, Guohao Huang, Xuan Chang, Bingni Wei, Yawang Sun, Zhengguo Yang, Yongju Zhao, Zhongquan Zhao, Guozhong Dong, Juncai Chen

**Affiliations:** 1College of Animal Science and Technology, Southwest University, Chongqing 400715, China; 15977756834@163.com (J.L.); huangguohao1997@163.com (G.H.); cx991103823@163.com (X.C.); 18294387678@163.com (B.W.); syaw507@swu.edu.cn (Y.S.); guoguo00002@163.com (Z.Y.); zyongju@163.com (Y.Z.); zhongquanzhao@swu.edu.cn (Z.Z.); gzdong@swu.edu.cn (G.D.); 2Chongqing Key Laboratory of Herbivore Science, Chongqing 400715, China

**Keywords:** MAC-T cells, transcriptome profiles, serotonylation, mammary gland involution, cell viability, tight junction

## Abstract

Serotonin (5-HT) has been reported to play an important role in mammary gland involution that is defined as the process through which the gland returns to a nonlactating state. However, the overall picture of the regulatory mechanisms of 5-HT and the effects of serotonylation on mammary gland involution still need to be further investigated. The current study aimed to investigate the effects of 5-HT on global gene expression profiles of bovine mammary epithelial cells (MAC-T) and to preliminarily examine whether the serotonylation involved in the mammary gland involution by using Monodansylcadaverine (MDC), a competitive inhibitor of transglutaminase 2. Results showed that a high concentration of 5-HT decreased viability and transepithelial electrical resistance (TEER) in MAC-T cells. Transcriptome analysis indicated that 2477 genes were differentially expressed in MAC-T cells treated with 200 μg/mL of 5-HT compared with the control group, and the Notch, p53, and PI3K-Akt signaling pathways were enriched. MDC influenced 5-HT-induced MAC-T cell death, fatty acid synthesis, and the formation and disruption of tight junctions. Overall, a high concentration of 5-HT is able to accelerate mammary gland involution, which may be regulated through the Notch, p53, and PI3K-Akt signaling pathways. Serotonylation is involved in bovine mammary gland involution.

## 1. Introduction

Mammary gland involution of dairy cows is a complex physiological process involving extensive cell apoptosis and proliferation that lead to renewal of the mammary epithelial cell population, which usually takes place at the end of a lactation period. The process of mammary gland involution actively starts within 2 d after the cessation of milk removal and seems to be complete after approximately 21 d [1]. This involution is characterized by a decrease in the synthesis and secretion of milk components, the turnover of mammary epithelial cells, the activation of various proteases, and an increase in tight junction permeability [2]. Once the involution of the mammary gland is complete, resistance to bacterial infection improves [3]. Due to the optimization of genetics and management, milk production in modern dairy cows is increasing dramatically, and a dairy cow can still produce 25 to 30 kg/d of milk at the time of dry-off [4]. The high milk secretion capacity delays the transition to the involute state, thereby increasing the risk of acquiring a new intramammary infection. Rajala-Schultz et al. reported that the risk of intramammary infection at calving increases by 77% for every 5 kg of milk produced above 12.5 kg at drying off [5]. Additionally, the accumulated milk in the udder at the beginning of the dry period increases the risk of milk leakage [6,7] and causes discomfort and pain to dairy cows [8]. Therefore, accelerating mammary gland involution, especially in modern, high-milk-producing dairy cows, would be beneficial to the health and welfare of the animals.

Serotonin (5-HT) is an indolamine compound synthesized from tryptophan that contains an indole ring and a carboxyl-amide side-chain [9]. It is well established that 5-HT plays an important role in triggering mammary gland involution through autocrine–paracrine mechanisms on mammary epithelial cells to regulate mammary gland development and milk secretion [10,11,12]. An increasing concentration of 5-HT was reported to promote mammary gland involution via the 5-HT7 receptor (HTR7) by disrupting tight junctions [13]. Field et al. administered intramammary infusions of serotonin precursor (5-hydroxy-l-tryptophan, 5-HTP) at drying-off and achieved a more efficient involution, reflected by increased cell death and epithelial tight junction permeability, and downregulation of milk protein genes [14]. However, to our knowledge, the effects of 5-HT on the mammary gland has not been studied using transcriptome analysis. In addition, recent studies have shown that 5-HT not only acts through receptors but may also play a physiological role through serotonylation after entering the cells [15]. Serotonylation is the covalent post-translational modification involving 5-HT, catalyzed by transglutaminases (TGs), especially by transglutaminase2 (TG2) [15]. Previous studies have revealed that TG2 regulates cell adhesion, cell signaling, and cell survival in various cell types [16,17,18]. Monodansylcadaverine (MDC) is a cell-permeable competitive inhibitor of TG2, which is usually used to inhibit TG2 activity [19]. However, the effects of serotonylation on mammary gland involution are still unclear. In addition, the MAC-T is one of the most often used immortalized bovine mammary epithelial cell lines to study bovine mammary gland functions, such as cell proliferation and apoptosis, tight junction permeability, inflammations, etc. [20,21,22,23]. Therefore, the objective of the current study was to investigate the effects of 5-HT on global gene expression profiles of bovine mammary epithelial cells by applying transcriptome analysis in MAC-T cells. Further, whether the serotonylation involved in the mammary gland involution was preliminarily examined using TG2 inhibitor MDC.

## 2. Results

### 2.1. Effects of 5-HT on Intracellular 5-HT Concentration in MAC-T Cells

Compared with the control group, the intracellular 5-HT concentrations were increased in cells treated with 0.2, 2, or 20 μg/mL of 5-HT for 24 h (*p* < 0.05, Figure 1). The cells treated with 2 μg/mL of 5-HT exhibited the greatest intracellular 5-HT concentration compared with other groups. There was no significant difference in intracellular 5-HT concentration between the control group and cells treated with 200 μg/mL of 5-HT (*p* > 0.05).

### 2.2. Effects of 5-HT on Cell Viability in MAC-T Cells

Cell Counting Kit 8 (CCK-8) assay showed that the cell viability was significantly increased after treatment with 0.2 μg/mL of 5-HT for 24 h (*p* < 0.05, Figure 2A). There was no significant difference in cell viability between the control group and cells treated with 2, 20, or 200 μg/mL of 5-HT for 24 h (*p* > 0.05). However, 200 μg/mL of 5-HT decreased cell viability more than 0.2, 2, or 20 μg/mL of 5-HT for 24 h (*p* < 0.05). There was no significant difference in cell viability between the control group and cells treated with 0.2, 2, or 20 μg/mL of 5-HT for 48 h (Figure 2B) and 72 h (Figure 2C) (*p* > 0.05), but the cell viability was significantly decreased after treatment with 200 μg/mL of 5-HT for 48 h and 72 h compared with the control group (*p* < 0.05).

### 2.3. Effects of 5-HT on Transepithelial Electrical Resistance (TEER) in MAC-T Cells

The MAC-T cells were cultured on Transwell permeable supports for 12 days prior to reaching plateau TEER (~6200 Ω cm^2^). Then, cells were treated with various concentrations of 5-HT, and 200 μg/mL of 5-HT caused a rapid decrease in TEER of MAC-T cell monolayer within 24 h (*p* < 0.05, Figure 3). However, 0.2, 2, or 20 μg/mL of 5-HT did not influence the TEER compared with the control group (*p* < 0.05).

### 2.4. Transcriptional Profile in the MAC-T Cells

Principal components analysis (PCA) plot showed that the first principal component (PC1) summarized 80.53% of the variability, whereas the second principal component (PC2) summarized 8.37% of the variability (Figure 4A). The cells treated with 200 μg/mL of 5-HT were clearly distinct from the other two groups (0, and 2 μg/mL of 5-HT) on PC1, whereas PC2 did not allow any distinction among groups. There were 71 genes that were significantly differentially expressed in cells treated with 2 μg/mL of 5-HT compared with the control group, including 36 up-regulated genes and 35 down-regulated genes. A total of 2477 genes were significantly differentially expressed in cells treated with 200 μg/mL of 5-HT compared with the control group, including 1105 up-regulated genes and 1372 down-regulated genes (Figure 4B). Venn diagram shows that there are 36 shared differentially expressed genes (DEGs) between the “200 vs. 0” and “2 vs. 0” comparisons. Totals of 2441 and 35 unique DEGs are in the “200 vs. 0” and “2 vs. 0” comparisons, respectively (Figure 4C).

### 2.5. Kyoto Encyclopedia of Genes and Genomes (KEGG) Enrichment Analysis

The results of KEGG enrichment analysis for “200 vs. 0” comparison showed that eight down-regulated DEGs (*CDH3*, *FN1, ITGB4, COL1A1, CDH2, CDH26, TG2, CLDN-8, and THBS1*) and one up-regulated DEG (*CDH15*) were enriched in biological adhesion (Figure 4D), seven down-regulated DEGs (*TGFB2, NOTCH2, TP53, TGFB1I1, AKT3, MMP9, and NOTCH1*) and one up-regulated DEG (*STAT1*) were enriched in regulation of signaling (Figure 4E), two down-regulated DEGs (*MYC* and *MET*) and three up-regulated DEGs (*CASP4*, *CASP7*, and *EGR1*) were enriched in cell death (Figure 4F), and three down-regulated DEGs (*ACLY, FASN, and ACACA*) were enriched in fatty acid metabolic process (Figure 4G).

Among the thirty six shared DEGs observed both in the “200 vs. 0” and “2 vs. 0”, seven down-regulated DEGs (*CLSTN2, FAM83C, NYX, BOLA-DRA, C6H4orf17, TFEC, and C19H17orf78*) and four up-regulated DEGs (*HOXA13, CATIP, KIF5C*, and *CYP1A1*) were observed (Figure 4H). As shown in Figure 4I, five down-regulated DEGs (*METRN, DZIP3, CDH4, MYO3B, and FAM221B*) and three up-regulated DEGs (*MEIOB, HSPB2, and S100A12*) were observed in “2 vs. 0”.

### 2.6. Gene Ontology (GO) Enrichment Analysis

The results of GO enrichment analysis for “200 vs. 0” comparison showed that eight specific pathways related to mammary gland involution were enriched, including the Notch signaling pathway (*DLL1, NOTCH2, NOTCH1,* and *JAG1*, Figure 5), p53 signaling pathway (*GADD45G, CHEK1, CCND1, GADD45B, SESN2, CDKN1A*, and *TP53*), steroid biosynthesis (*LSS, DHCR7, LIPA, SQLE, NSDHL*, and *DHCR24*), cellular senescence (*E2F2, MAPK3, AKT3, CDK6, CDKN1A, CDKN2B, MYC, TGFB2*, and *TP53*), cell adhesion molecules (*CLDN-6, NCAM1, SDC1, CDH2, SDC2, NCAM2, NEO1, CD274, CLDN-1, CLDN-8, SDC3, CD6, CDH3*, and *CDH15*), ferroptosis (*SLC7A11, CYBB, STEAP3, SAT1, FTL, ACSL5*, and *TP53*), PI3K-Akt signaling pathway (*PIK3AP1, MAPK3, BCL2L1, PIK3R5, AKT3, CDKN1A*, and *TP53*), and cell cycle (*CDK6, CDKN1A, CDKN2B, MYC, PLK1, TGFB2,* and *TP53*).

### 2.7. Effects of MDC and 5-HT on the mRNA Expressions of Selected Genes

Compared with the control group, 200 μg/mL of 5-HT and MDC increased the mRNA expression of *TGF-β1* (*p* < 0.05, Figure 6A), while 200 μg/mL 5-HT + MDC decreased the mRNA expression of *TGF-β1* compared to the 200 μg/mL 5-HT group (*p* < 0.05). The mRNA expression of *Caspase-3* was increased in the 200 μg/mL 5-HT group compared to the control group (*p* < 0.05), and the mRNA expression of *Caspase-3* was decreased in the 200 μg/mL 5-HT + MDC group compared to the 200 μg/mL 5-HT group (*p* < 0.05).

The mRNA expressions of *FASN* and *ACACA* were decreased in the 200 μg/mL 5-HT group compared with the control group (*p* < 0.05, Figure 6B). The mRNA expressions of *SCD-1* and FASN were increased in the MDC group compared with the control group (*p* < 0.05). In addition, the mRNA expression of *ACACA* was decreased in the 2 μg/mL 5-HT + MDC group compared with the 2 μg/mL 5-HT group (*p* < 0.05).

Compared with the control group, the mRNA expressions of *CLDN-4* was increased in the 2 μg/mL 5-HT group (*p* < 0.05, Figure 6C). The mRNA expressions of *CLDN-1* and *CLDN-8* were decreased in the 200 μg/mL 5-HT group compared to the control group (*p* < 0.05). The mRNA expressions of *CLDN-4, ZO-2*, and *E-cadherin* were increased (*p* < 0.05), and the mRNA expressions of *CLDN-1* and *CLDN-8* were decreased in the MDC group compared to the control group (*p* < 0.05). Compared with the 2 μg/mL 5-HT group, the mRNA expressions of *CLDN-1* and *CLDN-8* were decreased (*p* < 0.05), and the mRNA expression of *ZO-2* was increased in the 2 μg/mL 5-HT + MDC group (*p* < 0.05). Compared with the 200 μg/mL 5-HT group, the mRNA expressions of *CLDN-1, ZO-1*, and *E-cadherin* were increased (*p* < 0.05), and the mRNA expressions of *CLDN-4* and *ZO-2* were decreased in the 200 μg/mL 5-HT + MDC group (*p* < 0.05).

## 3. Discussion

Accelerating mammary gland involution during the dry period can reduce the incidence of milk leakage, thus reducing the risk of pathogen invasion and the incidence of intramammary infection [24,25]. It is well established that 5-HT is involved in mammary gland involution in late lactation [10]. Hernandez-Castellano et al. found that increasing 5-HT concentration in blood through i.v. infusion of 5-HT precursor 5-HTP may inhibit lactogenesis in bovines [26]. Also, Hernandez et al. reported that intramammary infusions of 5-HTP or selective 5-HT reuptake inhibitors caused an increase in the rate of milk decline in late lactation and accelerated mammary gland involution [27]. In the present study, we aim to elucidate the function of 5-HT on bovine mammary gland involution by treating MAC-T cells with 5-HT using transcriptomics.

Our results showed that long-time treatments (48 h or 72 h) with 200 μg/mL of 5-HT inhibited the proliferation of MAC-T cells. In addition, TEER is an indicator of epithelial barrier integrity, and the current study showed that 200 μg/mL of 5-HT led to a disruption of tight junctions of MAC-T cells. This result was in line with Hernandez et al. who observed that treatment of pBMEC with exogenous FLX resulted in a disassembly of tight junctions after 24 h [27]. These findings suggest that the high concentration of 5-HT could accelerate mammary gland involution, as indicated by a decrease in cell viability and disruption of tight junctions. Moreover, the low concentrations of 5-HT (0.2, 2, or 20 μg/mL) did not influence the tight junction of MAC-T cells in the present study. This result does not support previous findings that low concentrations of 5-HT promoted the formation of tight junctions [13]. One of the possible explanations for the inconsistency is the lower concentration of 5-HT (0.2, 2, or 20 μg/mL) used in the current study compared with that of the previous study (5 or 7.5 × 10^−4^ M) [13].

Transcriptomics analysis showed that gene expressions of *HOXA13, CATIP, KIF5C,* and *CYP1A1* were up-regulated, and the gene expressions of *CLSTN2, FAM83C, NYX*, *BOLA-DRA, C6H4orf17, TFEC*, and *C19H17orf78* were down-regulated in both 200 vs. 0 and 2 vs. 0. Manzella et al. found that 5-HT is a CYP1A1 substrate that competes with aryl hydrocarbon receptor (AhR) for CYP1A1 degradation, which could regulate tryptophan metabolism by promoting sustained AhR signaling [28]. The KIF5C is enriched in microtubules and is necessary for the transport of N-cadherin between the Golgi and the plasma membrane, facilitating cell-to-cell adhesion [29]. In addition, *BOLA-DRA* is directly related to the major histocompatibility complex (MHC) molecules, which promote antigen recognition and activation of immune responses, and polymorphisms in the *BOLA-DR* gene relate to the inhibition of mastitis [30]. The *FAM83C* family might be associated with increasing DNA copy numbers [31]. Thus, either low or high concentrations of 5-HT may regulate some common genes that are involved in tryptophan metabolism, cell adhesion, immune responses, and signal transduction.

The results of transcriptome analysis of the current study showed that 2441 genes were differentially expressed in the 200 vs. 0 group, including genes that were enriched in the biological adhesion, regulation of signaling, cell death, and the fatty acid metabolic process in response to high concentration of 5-HT treatment. In this study, the gene expressions of *Caspase-4, 7* were up-regulated, which was related to cell death. These results were consistent with Marti et al. who indicated that the activation of caspases participated in the pro-apoptosis signaling pathway, thereby involving mammary gland involution [32].

*FASN* and *ACACA* were related to the fatty acid metabolic process. Previous studies found that *FASN* [33] and *ACACA* [34] were involved in the de novo synthesis of fatty acids in mammary epithelial cells, which is one of the main sources of fatty acid in milk [35]. In the current study, the gene expressions of *FASN* and *ACACA* were down-regulated, indicating that the ability of MAC-T cells to synthesize fatty acid was decreased, which was consistent with Hurley, who reported that mammary gland involution resulted in the ability of mammary epithelial cells to synthesize milk fat being decreased [1].

The genes which were enriched in the biological adhesion, including *CLDN-6, CDH2, CLDN-8, CDH3*, and *CDH15*, play a central role in the barrier function of the epithelium. The CLDNs are a large family of membrane proteins whose classic function is to regulate the permeability of tight junctions in epithelia, and CLDN-8 is localized to the tight junctions, where it is presumably important in forming the paracellular barrier in epithelia during lactation [36]. Our results showed that *CLDN-8* was down-regulated, indicating that tight junction permeability was decreased, which was consistent with results of the TEER. In addition, ZOs are also reported to be involved in composition of tight junctions [37,38], and Field et al. found that 20 mg/teat 5-HTP intramammary infusions resulted in decreases in the mRNA expression of *ZO-1, ZO-2,* and *ZO-3* [14]. In the present study, however, the gene expressions of ZOs had no significant difference in the 200 vs. 0 group. This discrepancy could be due to the different form of 5-HT used in the current study.

Mammary gland involution is characterized by apoptosis of mammary epithelial cells [39], shedding of mammary epithelial cells [40], and reduction in milk component synthesis [41]. Our results show that most gene expressions related to cell death were up-regulated, and most gene expressions related to biological adhesion and fatty acid metabolic process were down-regulated, which were consistent with the mammary gland involution process. Pai et al. found that 5-HT accelerates mammary gland involution through the cAMP-PKA pathway [13]. In this study, the results of GO enrichment analysis suggested that the Notch signaling pathway may also play an important role in the promoting effect of 5-HT on mammary gland involution. Our results showed that the gene expressions of *NOTCH1* and *NOTCH2* were decreased. These genes have been demonstrated to play a biologically important role during cell development [42,43]. It had been shown that the Notch signaling pathway maintains epithelial cell homeostasis by regulating cell proliferation and differentiation, and NOTCH1 and NOTCH2 as the key Notch receptors controlling cell function [44]. Therefore, it can be hypothesized that a high concentration of 5-HT may inhibit cell proliferation and promote the degeneration of MAC-T cells by inhibiting the Notch signaling pathway. More studies will have to be conducted to determine the role of the Notch signaling pathway on mammary gland involution during treatment with 5-HT.

Dizeyi et al. found that 5-HT could induce the up-regulation of the PI3K-Akt signaling pathway to result in the promotion of cell proliferation [45]. In addition, the p53 signaling pathway has been shown to be involved in inducing apoptosis [46,47]. These previous studies were in agreement with the results of our GO enrichment analysis, which showed that the PI3K-Akt signaling pathway and the p53 signaling pathway were enriched, indicating that the PI3K-Akt signaling pathway and the p53 signaling pathway also play a key role in regulating mammary gland involution.

In recent years, various studies have gradually confirmed that 5-HT not only acted through the receptors but also might play a physiological role through serotonylation after entering the cells [15]. Therefore, in this study, we studied the effects of different concentrations of 5-HT treatments on intracellular 5-HT concentrations in MAC-T cells at first. Our results demonstrated that low concentrations of 5-HT (0.2, 2, or 20 μg/mL) increased intracellular 5-HT concentration and reached a peak level upon treatment with 2 μg/mL of 5-HT, indicating that 5-HT can be taken up into MAC-T cells. 5-HT transporter (SERT) is the main pathway for cells to actively take up 5-HT from extracellular fluid [48]. In the present study, however, the mRNAs for *SERT* were not identified by qPCR in MAC-T cells. It was reported that, in addition to SERT, 5-HT also can be taken up by other transporters, including the norepinephrine transporter (NET) [12,49], dopamine transporter (DAT) [12,50,51,52], organic cation transporters (OCT) [12], and plasma membrane monoamine transporter (PMAT) [12]. A previous study demonstrated that NET and DAT may play a more important role when extracellular levels of 5-HT are very high or when 5-HT transporter expression is eliminated [48]. Hence, it is possible that 5-HT can be taken up into MAC-T cells through other transporters. In addition, treatment with a higher concentration of 5-HT (200 μg/mL) did not cause significant changes in intracellular 5-HT concentration compared with the control group. This result might be explained by the fact that the high concentration of 5-HT caused apoptosis and decreased the capacity for 5-HT transportation in MAC-T cells.

Serotonylation is a newly recognized, post-translational modification with 5-HT, which is catalyzed by TGs, especially by TG2 [15]. TG2-mediated changes regulate multiple cellular responses, such as cell adhesion, cell signaling, and survival [16,17,18]. A previous study showed that 5-HT can increase TG2 expression [53]. MDC, as an alternative substrate inhibitor, can exploit the protein-crosslinking activity of TGs, thereby reducing serotonylation [19]. Guilluy et al. indicated that MDC could inhibit 5-HT-induced serotonylation and cell proliferation [54]. Also, Sheftel et al. treated mouse mammary epithelial cells with MDC and found that serotonylation was partly involved in the regulation of parathyroid hormone related-protein (PTHrP) by 5-HT under lactogenic conditions [19]. However, whether serotonylation plays a role in 5-HT-induced mammary gland involution is unclear. In this study, we treated MAC-T cells with MDC to inhibit serotonylation in order to elucidate the function of serotonylation in regulating gene expression.

TGF-β1-induced apoptosis and autophagy of MAC-T cells play an important role in bovine mammary gland involution [55,56]. Di et al. found that TGF-β1 can induce MAC-T cell apoptosis by stimulating Caspase-3 activation [57]. Our results showed that the mRNA expression of *TGF-β1* and *Caspase-3* was significantly inhibited by MDC in the high concentration of 5-HT group, indicating that a high concentration of 5-HT-induced MAC-T cell apoptosis was inhibited by MDC.

The SCD-1 [58], FASN [33], and ACACA [34] play an essential role in fatty acid synthesis. Our results showed that the mRNA expressions of *FASN* and *ACACA* were inhibited in the high concentration of the 5-HT group, demonstrating that the ability of fatty acid synthesis was decreased, which was consistent with the results of our KEGG pathway analysis. However, the current study showed that MDC had no effect on the mRNA expressions of *SCD-1, FASN,* and *ACACA* in MAC-T cells, which were treated with a high concentration of 5-HT. Thus, the high concentration of 5-HT inhibited gene expressions related to fatty acid synthesis, especially *FASN* and *ACACA*, but the inhibitory effect of a high concentration of 5-HT may not be influenced by serotonylation. It should be noted that, compared with the control group, the mRNA expression of *SCD-1* and *FASN* were up-regulated in the MDC group in this study. But there are few studies about the effect of serotonylation on fatty acid synthesis, and the mechanism remains unclear.

It is well established that CLDNs [36] and ZOs [37,38] participate in the formation of tight junctions in epithelial cells. Also, E-cadherin is the main transmembrane adherent junction protein expressed in luminal cells in the mammary gland [59]. Our results showed that a high concentration of 5-HT significantly inhibited the mRNA expression of *CLDN-1* and *CLDN-8*, which were consistent with the results of our KEGG pathway analysis. In addition, MDC treatment resulted in the up-regulation of the mRNA expression of *CLDN-1, ZO-1*, and *E-cadherin* when cells were treated with a high concentration of 5-HT, but MDC had no significant effects on *CLDN-8* and *ZO-2*. Our results also showed that MDC significantly down-regulated the mRNA expressions of *CLDN-1* and *CLDN-8*, and up-regulated the mRNA expression of *ZO-2* in cells treated with a low concentration of 5-HT. These results demonstrated that serotonylation may be involved not only in the high concentration of 5-HT-induced disruption of tight junctions through the regulation of gene expression of *CLDN-1, ZO-1*, and *E-cadherin*, but also in low concentration of 5-HT-induced formation of tight junctions through the regulation of genes expression of *CLDN-1, CLDN-8,* and *ZO-2* in MAC-T cells.

Taken together, we speculated that serotonylation may be involved in 5-HT-induced cell death, the regulation of fatty acid synthesis, and the formation and disruption of tight junctions via regulating the expression of related genes. However, the molecular mechanism involved in accelerating mammary gland involution through serotonylation remains to be further explored.

## 4. Materials and Methods

### 4.1. Cell Culture

Bovine mammary epithelial cells (MAC-T), an immortalized bovine mammary epithelial cell line, were resuscitated and cultured in DMEM/F12 supplemented with 10% FBS and 1% streptomycin/penicillin in T-25 cell culture flasks (Thermo Fisher, Waltham, MA, USA) in a humidified atmosphere of 5% CO_2_ at 37 °C. The media was renewed every two days, and confluent layers of cells were detached with trypsin and replated into new plates at a ratio of 1:2 for the subsequent experiments when the MAC-T cell monolayer reached 80–90% confluency. Cell washing with PBS was performed before each renewal of media and passage.

### 4.2. Cell Treatment

The MAC-T cells were cultured in DMEM/F12 supplemented with 10% FBS and 1% streptomycin/penicillin for 24 h, and the mediums were removed. The cells were then incubated in mediums with different concentrations of 5-HT (0, 0.2, 2, 20, or 200 μg/mL) for the indicated times. Finally, the cells were routinely harvested for the subsequent experiments.

For the qPCR analysis, MAC-T cells were incubated in treatment mediums with different concentrations of 5-HT treatments and MDC (control, 2 μg/mL 5-HT, 200 μg/mL 5-HT, 200 µM MDC, 2 μg/mL 5-HT + 200 µM MDC, or 200 μg/mL 5-HT + 200 µM MDC) for 24 h.

### 4.3. Intracellular 5-HT Concentrations

The MAC-T cells were washed twice using PBS and were lysed with lysis buffer containing 1% deoxycholate, 1% Triton X-100, and 0.1% SDS for 15 min at 4 °C. Protein concentration was analyzed using a BCA protein assay kit according to the manufacturer’s instructions (Beijing Solarbio Co., Ltd., Beijing, China). The intracellular 5-HT concentrations were determined via bovine enzyme-linked immunosorbent assay (ELISA) Kit (YX-050820B, Sino Best Biological Technology Co., Ltd., Shanghai, China) according to the manufacturer’s instructions. All assays had an intraassay CV of <10% and an interassay CV of <5%.

### 4.4. Cell Viability Assay

Cell viability was determined using Cell Counting Kit 8 (CCK-8) according to the manufacturer’s protocol (Beijing Solarbio Co., Ltd., Beijing, China). MAC-T cells were treated with different concentrations of 5-HT (0, 0.2, 2, 20, or 200 μg/mL) in a humidified atmosphere of 5% CO_2_ at 37 °C for 24, 48, or 72 h. The absorbance was tested via Microplate reader (Thermo, Varioskan LUX, Waltham, MA, USA) at 450 nm. Cell viability was expressed as the percentage of control.

### 4.5. Transepithelial Electrical Resistance (TEER) Measurement

The tight junction permeability of the MAC-T cell monolayer was evaluated by measuring TEER. The MAC-T cells were seeded in DMEM/F12 supplemented with 10% FBS and 1% streptomycin/penicillin on clear polyester permeable supports (Transwell, Corning; 0.4 mm pores, polyester). Mediums were changed every 24 h. Resistance measurements were taken daily. The indicated concentrations of 5-HT (0, 0.2, 2, 20, or 200 μg/mL) were added to the culture medium when TEER reached a plateau. The pure cell culture medium without MAC-T cells was used as a blank control. The blank resistance (which is usually ~30 Ω/cm^2^) was then subtracted from the measured resistance to obtain the effective TEER.

### 4.6. RNA Extraction and Sequencing

MAC-T cells were treated with different concentrations of 5-HT (0, 2, or 200 μg/mL) for 24 h and were then collected for RNA-Seq. Total RNA (20 μg) was extracted from MAC-T cells by adding 0.4 mL of Trizol reagent (TIANGEN BIOTECH, Beijing, China) per 1 × 10^5^–10^7^ cells directly into the cell culture dish according to the manufacturer’s protocol. RNA degradation and contamination were monitored on 1% agarose gels. RNA concentration and purity were examined using a NanoDrop spectrophotometer (NanoDrop Technologies, Wilmington, DE, USA). RNA integrity was assessed using the Agilent 2100 Bioanalyzer (Agilent Technologies, Santa Clara, CA, USA). A total of 9 RNA samples [3 samples per group × 3 groups (0, 2, or 200 μg/mL of 5-HT)] were sent to Allwegene Technology Co., Ltd. (Beijing, China) for cDNA library construction and sequencing. Briefly, the library was constructed using the NEBNext UltraTM RNA Library Prep Kit for Illumina (NEB) following the manufacturer’s manual. The samples were sequenced on the Illumina HiSeq-X platform (Illumina, SanDiego, CA, USA) and paired-end reads were generated.

### 4.7. Library Preparation

A total amount of 1.5 μg RNA per sample was used as input material for the RNA sample preparations. Sequencing libraries were generated using NEBNext^®^ UltraTM RNA Library Prep Kit for Illumina^®^ (NEB, Nebraska, NE, USA), following the manufacturer’s recommendations, and index codes were added to attribute sequences to each sample. A total of 403,657,472 raw reads were generated. The library preparations were sequenced on an Illumina Novaseq 6000 platform by the Beijing Allwegene Technology Company Limited (Beijing, China) and paired-end 150 bp reads were generated.

### 4.8. Data Processing and Differentially Expressed Gene Screening

Raw data of fastq format were initially processed through in-house perl scripts. In this step, clean reads were obtained by removing reads containing adapter, reads containing ploy-N, and low-quality reads from raw data. Q20, Q30, and GC-content were calculated for the clean data. These clean data were then mapped to the reference genome sequence using STAR. All the downstream analyses were based on clean data with high quality. HTSeq v 0.5.4 p3 was used to count the reads numbers mapped to each gene. Gene expression levels were estimated from fragments per kilobase of transcript per million fragments mapped (FPKM).

The differential expressed gene among the three groups (three biological replicates per group) was analyzed using DESeq R package (1.10.1). DESeq provides statistical routines for determining differential expression in digital gene expression data using a model based on the negative binomial distribution. The resulting *p* values were adjusted using the Benjamini and Hochberg’s approach for controlling the false discovery rate. Genes with an adjusted *p*-value < 0.05 found using DESeq were assigned as differentially expressed.

### 4.9. Quantitative Real-Time PCR (qPCR)

The expression of genes *CLDN-1, CLDN-4, CLDN-8, TJP1, TJP2, CDH1, TGF-β1, CASP3, SCD-1, FASN*, and *ACACA* were detected using quantitative real-time PCR (RT-qPCR). Briefly, total RNA was extracted using TRIzol reagent (Thermo Fisher Scientific) according to the manufacturer’s instructions. After the synthesis of first-strand cDNA from 1 μg of RNA using the PrimeScript RT Reagent Kit (TaKaRa, Dalian, China), qPCR was performed using TaKaRa SYBR^®^ Premix Ex Taq™ II (Perfect Real Time) on a LightCycler^®^ 480 system (Roche Diagnostics). The reactions were incubated at 95 °C for 30 s, followed by 40 cycles of 95 °C for 5 s, 55 °C for 30 s, and 72 °C for 30 s. RPS-9 and GAPDH were used as the endogenous reference genes. All reactions were run in three replicates for each sample. The primer sequences were listed in Appendix A. Analysis was conducted using the 2^−∆∆CT^ method.

### 4.10. Statistical Analysis

Statistical analysis was performed via one-way ANOVA with Tukey’s post hoc tests using GraphPad Prism 9 for macOS (Version 9.2.0, GraphPad Software, San Diego, CA, USA, www.graphpad.com, accessed on 11 November 2022). Sankey dot of GO enrichment analyses were mapped using a free online platform for data analysis and visualization (https://www.bioinformatics.com.cn, accessed on 11 November 2022). The results are presented as means with standard error of the means (SEM). The variant letter indicates a significant difference (*p* < 0.05).

## 5. Conclusions

In summary, our study demonstrated that the effect of 5-HT on MAC-T cell viability and intercellular tight junctions is concentration dependent. A low concentration of 5-HT promotes MAC-T cell proliferation, but a high concentration of 5-HT inhibits MAC-T cell proliferation and causes disruption of tight junctions, thus accelerating mammary gland involution. A high concentration of 5-HT may promote bovine mammary gland involution by regulating the Notch, p53, and PI3K-Akt signaling pathways. Serotonylation is involved in 5-HT-induced MAC-T cell death, fatty acid synthesis, and the formation and disruption of tight junctions, thereby regulating bovine mammary gland involution.

## Figures and Tables

**Figure 1 ijms-24-11388-f001:**
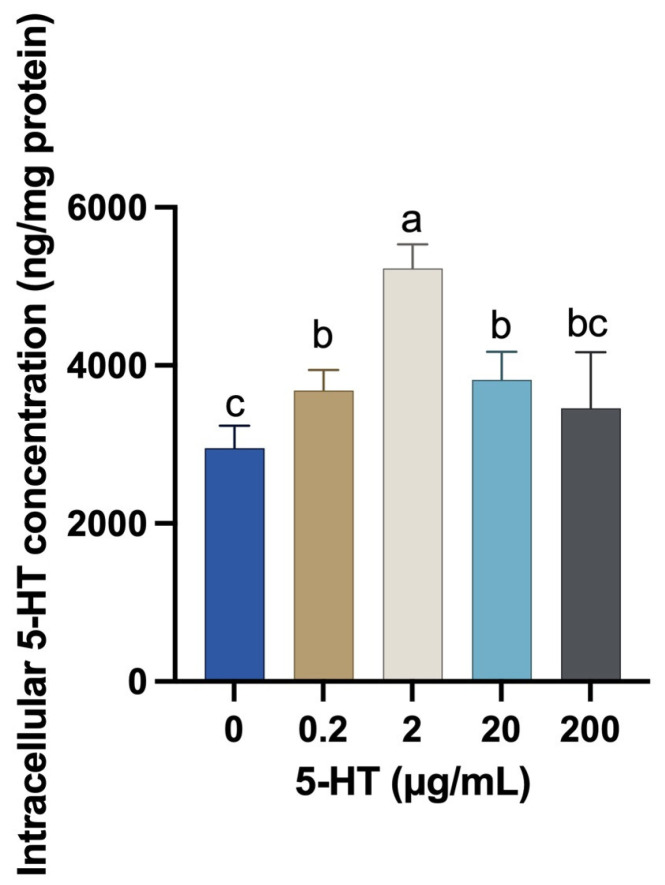
Effects of 5-HT on intracellular 5-HT concentrations in MAC-T cells. MAC-T cells were treated with the indicated concentrations of 5-HT (0, 0.2, 2, 20, or 200 μg/mL) for 24 h. The values represent the mean ± SEM (*n* = 6). The variant letter indicates a significant difference when *p* < 0.05.

**Figure 2 ijms-24-11388-f002:**
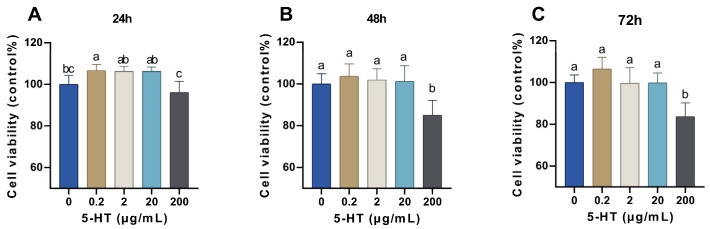
Effects of 5-HT on MAC-T cell activity. (**A**–**C**) MAC-T cells were treated with the indicated concentrations of 5-HT (0, 0.2, 2, 20, or 200 μg/mL) for the indicated times (24, 48, or 72 h). The values represent the mean ± SEM (*n* = 6). The variant letter indicates a significant difference when *p* < 0.05.

**Figure 3 ijms-24-11388-f003:**
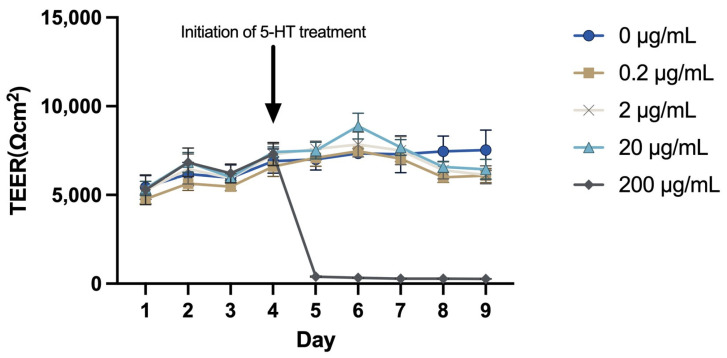
Effects of 5-HT on transepithelial electrical resistance (TEER) in MAC-T cells (Ωcm^2^). The values represent the mean ± SEM (*n* = 6). The arrow indicates the time-point when indicated concentrations of 5-HT (0, 0.2, 2, 20, or 200 μg/mL) was added.

**Figure 4 ijms-24-11388-f004:**
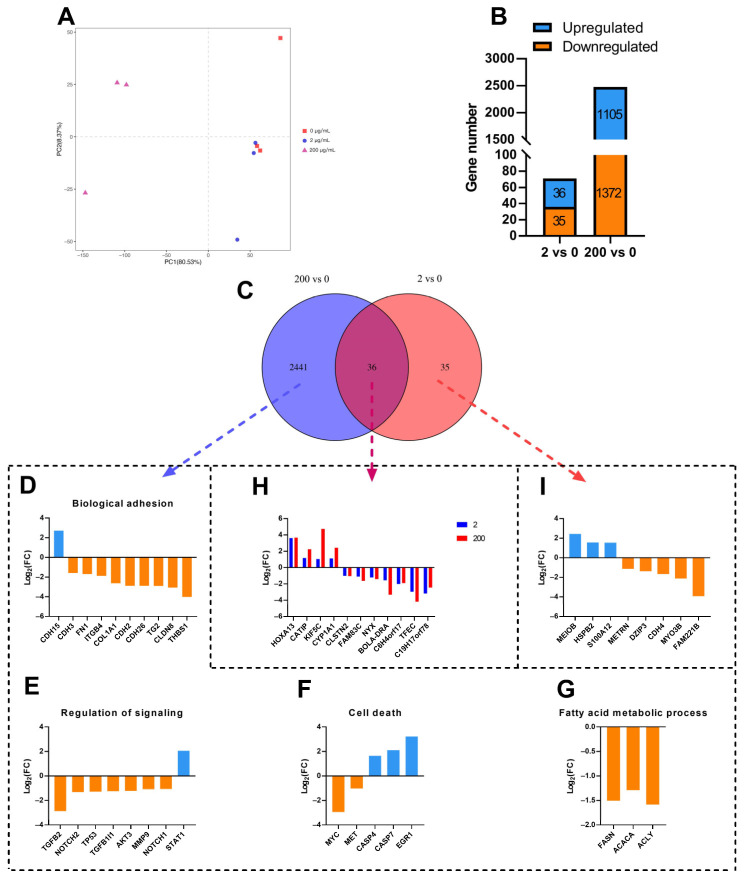
PCA analysis of gene expressions, number of differentially expressed genes, and main differentially expressed genes in MAC-T cells treated with 0, 2, or 200 μg/mL of 5-HT. (**A**) The PCA analysis of gene expressions in MAC-T cells among the three groups. (**B**) The number of differentially expressed genes in “2 vs. 0” and “200 vs. 0”. (**C**) Venn diagram of the altered gene expressions among three groups. The number of unique and shared expression genes in “2 vs. 0” and “200 vs. 0” were indicated. (**D**–**G**) The KEGG pathways significantly enriched in the unique differentially expressed genes identified in MAC-T cells in “200 vs. 0”, and differentially expressed genes in main KEGG pathways. (**H**) The shared differentially expressed genes identified in MAC-T cells between “200 vs. 0” and “2 vs. 0”. (**I**) The unique differentially expressed genes identified in MAC-T cells in “2 vs. 0”.

**Figure 5 ijms-24-11388-f005:**
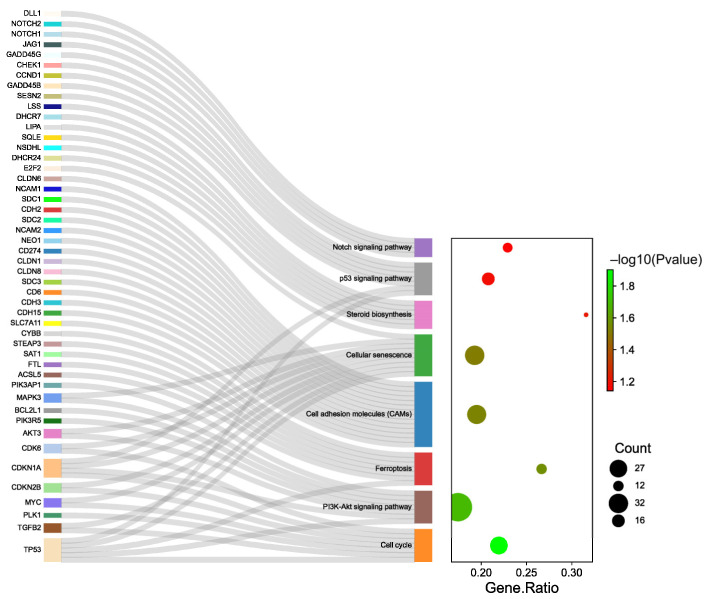
Summary of GO enrichment analysis of the unique expression genes in MAC-T cells of 200 vs. 0. The differentially expressed genes in 8 specific pathways.

**Figure 6 ijms-24-11388-f006:**
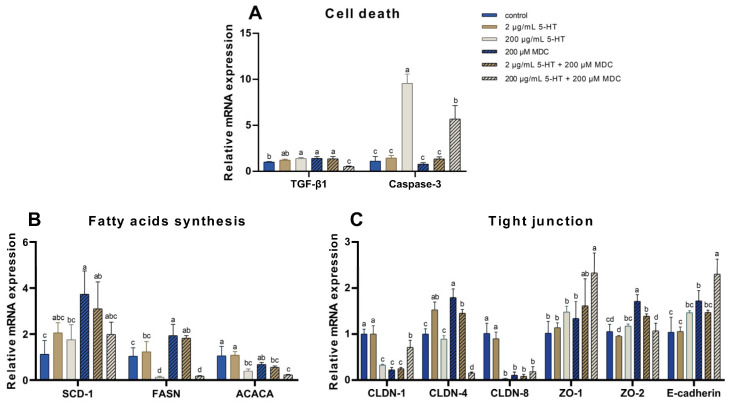
Effects of 5-HT (0, 2, or 200 μg/mL) and 200 μM monodansylcadaverine (MDC) on the mRNA expressions of selected genes. (**A**) RT-PCR analyses of the relative mRNA expression of genes associated with cell death. (**B**) RT-PCR analyses of mRNA expression of genes associated with fatty acids synthesis. (**C**) RT-PCR analyses of mRNA expression of genes associated with tight junction. The values represent the mean ± SEM (*n* = 6). The variant letter in the same bar chart indicates a significant difference when *p* < 0.05.

## Data Availability

The datasets presented in this study can be found in online repositories: https://www.ncbi.nlm.nih.gov/bioproject/PRJNA977548 (accessed on 31 May 2023).

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
