# Peer review of "Effects of Serotonin on Cell Viability, Permeability of Bovine Mammary Gland Epithelial Cells and Their Transcriptome Analysis"

_ijms, 2023, doi:10.3390/ijms241411388_

Round 1

Reviewer 1 Report

Congrats on a great job.

Just a little observation: the word vs must be written in italics and without dot.

Author Response

Just a little observation: the word vs must be written in italics and without dot.

Reply: Thank you very much, and we’ve revised it.

Reviewer 2 Report

The English language is fine. The introduction should contain more of the background to support the study.

Author Response

Major comment

The authors showed interesting results on the effects of different serotonin concentrations of 5-HT on bovine mammary epithelial cells in culture. High concentration of 5-HT (200 ug/ml) induced a reduction in cell viability and reduction of membrane integrity and no effect on the concentration of cell serotonin. How far is this concentration of 5-HT from a physiological level that would affect cells in vivo? The authors completed the study with a transcriptomic analysis of the genes expressed with different concentration of 5-HT and blocked transglutaminase to analyse serotonylation. 

The abstract and introduction should be improved to avoid repetitions and to better explain what is known, which methods are used so that a person outside the field could follow the paper. The cell specificity should be better described, are they primary cells, transformed cells, how do they recapitulate the mammary gland. If they are primary cells, could the effect of the high 5-HT concentration due to early senescence? For the transcriptomic analysis details on the differential expression analysis are totally missing in the methods, which methods, how many reads, how was the library preprared, which methods were used for the differential expressed analysis? A weighted gene co-expression network analysis maybe more informative and global than the KEGG analysis, which appears to focus on few genes, or it should be better explained what was done.

In the discussion the results of figure 1 should be discussed. The fluidity of the discussion should be improved. The authors started with infection to cell viability to transcriptome to serotonylation to TGF-b...

Reply: Thanks a lot for the thoroughly reviewing that could significantly improve the quality of this manuscript. The physiological 5-HT level in vivo is relatively lower that the greatest 5-HT concertation used in this study (200 μg/mL). The range of 5-HT concentration used in the current study was set by reference to the study of Pai et al., (2008, The Journal of Biological Chemistry, 283, 45, 30901-30910). The in vivo circumstances are much more complicated than in vitro, which may explain the lower 5-HT concentration could influence the cells in vivo.

We tried to improve the clarity of abstract and introduction. The MAC-T is one of the most often used immortalized bovine mammary epithelial cell lines to study the bovine mammary gland functions (L69-71). The SA-β-galactosidase is the biomarker of cell early senescence. We’ve checked the transcriptome data, and the change of relevant mRNA was not detected. So, we think it is less likely to be cell early senescent.

 In addition, the details on differential expression analysis were added (L419-457).

We did not use weighted gene co-expression network analysis in the current study due to the limited sample numbers (9). We may use it in the future study with larger sample size.

We discussed the figure 1 in the discussion and tried to improve the fluidity of the discussion.

Minor comment

Abstract:

L11-13: The two sentences can be merged, and involution should be defined.

Reply: We added the definition of involution in the abstract. (L12)

L15+23: Serotonylation is the covalent linkage of serotine to proteins, thus this term may not be what the authors want to do in this study. If you mean this, you should explain it.

Reply: Thanks a lot for pointing out this improper statement. Actually, the Serotonylation is the covalent post-translational modification involving 5-HT. We’ve changed the definition in the introduction.

L17: What is TEER?

Reply: TEER is transepithelial electrical resistance. Its full name was added (L18).

L20: What is MDC?

Reply: MDC is Monodansylcadaverine. Its full name and a brief introduction to it were added in abstract (L21-22).

L22-23: Is serotonin involved in cell senescence, apoptosis? What is the difference between cell senescence and involution and apoptosis?

Reply: So far, it is clear that serotonin is involved in cell apoptosis, and may be involved in cell senescence. Cell senescence is a process characterized by irreversible cell-cycle arrest, resistance to apoptosis, phenotypic remodeling and the release of bioactive factors (cytokines, chemokines and proteases). The mammary gland involution is characterized by a decrease in the synthesis and secretion of milk components, the turnover of mammary epithelial cell (apoptosis and proliferation), the activation of various proteases, and an increase in tight junction permeability. Apoptosis is a programmed form of cell death. Since the apoptosis and senescence

The SA-β-galactosidase is the biomarker of cell early senescence. We’ve checked the transcriptome data, and the change of relevant mRNA was not detected. So, we think it is less likely to be cell early senescent.

Introduction:

The introduction could be longer.

Explain better the process of involution: what are the mechanisms: eg apoptosis, tight junction degradation…, the effect of serotonin on mammary gland, serotonylation, what is known and then why the authors’ experiment cell system could recapitulate these steps and what they will show.

Reply: We added more information about the mammary gland involution, effects of serotonin on mammary gland.(L32-36;L52-55ï¼›L61-68)

Results:

L74: what is a CCK-8 assay

Reply: CCK-8 assay is Cell Counting Kit 8 (CCK-8) assay. Its full name was added.

L108: what do the authors mean with the KEGG? Is it a pathway enrichment analysis?

Reply: Yes, KEGG is Kyoto Encyclopedia of Genes and Genomes. Its full name was added. It is a pathway enrichment analysis.

2.1 Effect of 5HT on the intracellular level of 5HT, why an effect only with 2 ug/ml and not the other one: is it a timing problem?

Reply: We actually expected a dosage effects which the highest 5-HT concentration had greatest intracellular level of 5-HT. The possible explanation could be that high concentration of 5-HT caused apoptosis and decreased the capacity of 5-HT transportation in MAC-T cells. We added this in the discussion (L305-307)

Discussion

How are the results of figure 1 explained?

Reply:The high concentration of 5-HT might apoptosis and decreased the capacity of 5-HT transportation in MAC-T cells. We added this in the discussion (L305-307)

L247-249 would fit in the introduction to better explain the mechanism at play in the mammary glands.

Reply:Thanks for this suggestion and the characteristics of the mammary gland involution is introduced in the introduction to better explain the mechanism (L32-36).

Methods:

L350: MAC-T cells are they primary cells or a cell line? If they are primary cells, does 5-HT induce senescence?

Reply: MAC-T is an immortalized bovine mammary epithelial cell line. We added the details of MAC-T for more clarity (L360-361). The SA-β-galactosidase is the biomarker of cell early senescence. We’ve checked the transcriptome data, and the change of relevant mRNA was not detected. So, we think it is less likely to be cell early senescent.

L361: what is the “treatment medium”.

Reply: Thanks and we corrected it.

L373: what is CCK-8?

Reply: CCK-8 assay is Cell Counting Kit 8 (CCK-8) assay. Its full name was added.

L393: RNA-seq: how many reads were obtained, to what where they mapped, how were the differential gene found, which methods? Was there a correction for multiple testing?

Reply: A total of 403,657,472 raw reads were generated. The data were then mapped to the reference genome sequence by STAR. The details of the data processing and DEG screening were added  .The multiple testing was corrected by Benjamini and Hochberg’s approach. The above information were all added in M&M (L417-440).

Reviewer 3 Report

I reviewed the manuscript entitled “Effects of serotonin on cell viability, permeability of bovine mammary gland epithelial cells and their transcriptome analysis”

Abstract

-Line 17: please add full name of TEER

-Line 20: please add full name of MDC and some information about its function related to serotonin

Introduction

The first part of introduction is described sufficiently and introduces the reader to the importance of mammary gland involution related to milk production. Nevertheless, in the second paragraph I would have liked to read more about serotonylation and transglutaminases, especially MDC.

Results

The results are presented logically and allow the reader to follow the analysis course. I have a couple suggestions and questions:

-Line 70: “Effects of 5-HT”

-Line 81: “for 48 h and 72 h”

-Line 98: Please add more explanation about the result of PCA analysis. This one sentence is not enough.

-Line 110: “down-regulated differentially expressed genes (DEGs)”

-Figure 6: y-axis of the charts: Relative mRNA expression

-Figure 6: the names of the treatments are not clear. The color explanation is misleading (0, 2, 200, MDC, 2 MDC ?). The Figure 6 should follow the names of groups in the Materials and Methods, where the groups are control, 2 μg/mL 5-HT, 200 μg/mL 5-HT, 200 μ M MDC, 2 μg/mL 5-HT + 200 μM MDC, or 200 μg/mL 5-HT + 200 μ M MDC.

The authors discussed the research results concerning the literature. The discussion is very well detailed, understandable to the reader.

Materials and Methods

-Line 342-348: not necessary to specify in a separate paragraph the manufacturer of the materials, it is sufficient mention in the text.

-Line 350: the names of the professors could be included in the acknowledgements instead of the materials and methods.

-Line 358: in the gene expression experiment were applied different cell treatments (MDC), it would be worth to mention in this paragraph.

-Line 373: Cell Counting Kit 8 (CCK-8)

-Line 391: Could you briefly add some information about protocol of RNA isolation?

-Line 391: How did you checked the RNA integrity after isolation?

-Line 415: Please provide the list of target genes

-Line 415: How many reference genes did you test? How did you choose the most stable reference genes? Why did you used two different reference genes?

Author Response

I reviewed the manuscript entitled “Effects of serotonin on cell viability, permeability of bovine mammary gland epithelial cells and their transcriptome analysis”

Abstract

-Line 17: please add full name of TEER

Reply: Thanks, and we added the full name of TEER (L18).

-Line 20: please add full name of MDC and some information about its function related to serotonin

Reply: Thanks, and we added the full name of MDC and indicated it’s funtion (L21-22).

Introduction

The first part of introduction is described sufficiently and introduces the reader to the importance of mammary gland involution related to milk production. Nevertheless, in the second paragraph I would have liked to read more about serotonylation and transglutaminases, especially MDC.

Reply: Thanks for this suggestion, and we added some extra information about serotonylation, transglutaminases, and MDC (L61-68).

Results

The results are presented logically and allow the reader to follow the analysis course. I have a couple suggestions and questions:

-Line 70: “Effects of 5-HT”

Reply: Addressed.

-Line 81: “for 48 h and 72 h”

Reply: Addressed.

-Line 98: Please add more explanation about the result of PCA analysis. This one sentence is not enough.

Reply: We added more explanation about the PCA analysis. (L113-L117)

-Line 110: “down-regulated differentially expressed genes (DEGs)”

Reply: Addressed.

-Figure 6: y-axis of the charts: Relative mRNA expression

Reply: Addressed.

-Figure 6: the names of the treatments are not clear. The color explanation is misleading (0, 2, 200, MDC, 2 MDC ?). The Figure 6 should follow the names of groups in the Materials and Methods, where the groups are control, 2 μg/mL 5-HT, 200 μg/mL 5-HT, 200 μ M MDC, 2 μg/mL 5-HT + 200 μM MDC, or 200 μg/mL 5-HT + 200 μ M MDC.

Reply: Indeed, the names of the treatments are not accurate and clear enough. Thanks for the kind reminder and we revised them.

The authors discussed the research results concerning the literature. The discussion is very well detailed, understandable to the reader.

Materials and Methods

-Line 342-348: not necessary to specify in a separate paragraph the manufacturer of the materials, it is sufficient mention in the text.

Reply: Thanks for this nice suggestion, we’ve removed it.

-Line 350: the names of the professors could be included in the acknowledgements instead of the materials and methods.

Reply: We moved names to the acknowledgements.

-Line 358: in the gene expression experiment were applied different cell treatments (MDC), it would be worth to mention in this paragraph.

Reply: Thanks a lot. We mentioned it in that paragraph (L373-376).

-Line 373: Cell Counting Kit 8 (CCK-8)

Reply: Addressed.

-Line 391: Could you briefly add some information about protocol of RNA isolation?

Reply: Yes, we briefly described the protocol of RNA isolation.

-Line 391: How did you checked the RNA integrity after isolation?

Reply:The RNA integrity was monitored by Agilent 2100 Bioanalyzer. We added it in the text. (L409-410)

-Line 415: Please provide the list of target genes

Reply: Thanks and we provided the list or target genes.

-Line 415: How many reference genes did you test? How did you choose the most stable reference genes? Why did you used two different reference genes?

Reply: we have tested two reference genes, RPS9 and GAPDH, which are widely used as housekeeping genes for MAC-T cells study. These two genes are both stable and we calculated the geometric mean of the two housekeeping genes to get a more reliable result.